# COVID-19—The Potential Beneficial Therapeutic Effects of Spironolactone during SARS-CoV-2 Infection

**DOI:** 10.3390/ph14010071

**Published:** 2021-01-17

**Authors:** Katarzyna Kotfis, Kacper Lechowicz, Sylwester Drożdżal, Paulina Niedźwiedzka-Rystwej, Tomasz K. Wojdacz, Ewelina Grywalska, Jowita Biernawska, Magda Wiśniewska, Miłosz Parczewski

**Affiliations:** 1Department of Anesthesiology, Intensive Therapy and Acute Intoxications, Pomeranian Medical University in Szczecin, 70-111 Szczecin, Poland; kacper.lechowicz@pum.edu.pl; 2Department of Pharmacokinetics and Monitored Therapy, Pomeranian Medical University, 70-111 Szczecin, Poland; starkdrozd@wp.pl; 3Institute of Biology, University of Szczecin, 71-412 Szczecin, Poland; paulina.niedzwiedzka-rystwej@usz.edu.pl; 4Independent Clinical Epigenetics Laboratory, Pomeranian Medical University, 71-252 Szczecin, Poland; tomasz.wojdacz@pum.edu.pl; 5Department of Clinical Immunology and Immunotherapy, Medical University of Lublin, 20-093 Lublin, Poland; ewelina.grywalska@gmail.com; 6Department of Anesthesiology and Intensive Therapy, Pomeranian Medical University in Szczecin, 71-252 Szczecin, Poland; jowita.biernawska@pum.edu.pl; 7Clinical Department of Nephrology, Transplantology and Internal Medicine, Pomeranian Medical University, 70-111 Szczecin, Poland; mwisniewska35@gmail.com; 8Department of Infectious, Tropical Diseases and Immune Deficiency, Pomeranian Medical University in Szczecin, 71-455 Szczecin, Poland; mparczewski@yahoo.co.uk

**Keywords:** COVID-19, SARS-CoV-2, coronavirus, pandemic, spironolactone, potassium canrenoate, ARDS (acute respiratory distress syndrome), androgen receptor antagonist, TMPRSS2

## Abstract

In March 2020, coronavirus disease 2019 (COVID-19) caused by SARS-CoV-2 was declared a global pandemic by the World Health Organization (WHO). The clinical course of the disease is unpredictable but may lead to severe acute respiratory infection (SARI) and pneumonia leading to acute respiratory distress syndrome (ARDS). It has been shown that pulmonary fibrosis may be one of the major long-term complications of COVID-19. In animal models, the use of spironolactone was proven to be an important drug in the prevention of pulmonary fibrosis. Through its dual action as a mineralocorticoid receptor (MR) antagonist and an androgenic inhibitor, spironolactone can provide significant benefits concerning COVID-19 infection. The primary effect of spironolactone in reducing pulmonary edema may also be beneficial in COVID-19 ARDS. Spironolactone is a well-known, widely used and safe anti-hypertensive and antiandrogenic medication. It has potassium-sparing diuretic action by antagonizing mineralocorticoid receptors (MRs). Spironolactone and potassium canrenoate, exerting combined pleiotropic action, may provide a therapeutic benefit to patients with COVID-19 pneumonia through antiandrogen, MR blocking, antifibrotic and anti-hyperinflammatory action. It has been proposed that spironolactone may prevent acute lung injury in COVID-19 infection due to its pleiotropic effects with favorable renin–angiotensin–aldosterone system (RAAS) and ACE2 expression, reduction in transmembrane serine protease 2 (TMPRSS2) activity and antiandrogenic action, and therefore it may prove to act as additional protection for patients at highest risk of severe pneumonia. Future prospective clinical trials are warranted to evaluate its therapeutic potential.

## 1. Introduction

In March 2020, coronavirus disease 2019 (COVID-19) caused by SARS-CoV-2 was declared a global pandemic by the World Health Organization (WHO), deeming it a threat to health and life [1]. Patients with SARS-CoV-2 infection include both asymptomatic carriers and those suffering from severe acute respiratory infection (SARI) leading to acute respiratory distress syndrome (ARDS), which may be complicated by sepsis and death. The clinical course of the disease is unpredictable but has been divided into certain stages, based on severity: 1. asymptomatic patient/mild course (mild upper or genitourinary symptoms); 2. stable patients (respiratory symptoms, radiological pneumonia); 3. unstable patients (respiratory failure); 4. ARDS patients (shock, multiorgan failure, impaired consciousness) [2].

In connection with the global SARS-CoV-2 infection outbreak that began in December 2019, spironolactone has become a light of potential importance in the treatment of complications in patients with COVID-19 infection [3]. It has been shown in animal models that the use of spironolactone may be important in the prevention of pulmonary fibrosis [4]. In addition, it has been shown that pulmonary fibrosis may be one of the major long-term complications of COVID-19 that might leave many patients debilitated and unable to return to pre-infection pulmonary fitness ^5^. Through its dual action as a mineralocorticoid receptor (MR) antagonist and an androgenic inhibitor, spironolactone can provide significant benefits in the treatment of one of the complications of COVID-19, that is, ARDS. In addition, the primary effect of spironolactone in reducing pulmonary edema may also be beneficial in COVID-19 ARDS [5]. Spironolactone is a well-known, widely used and safe anti-hypertensive and antiandrogenic medication. It has potassium-sparing diuretic action by antagonizing mineralocorticoid receptors (MRs). It has been proposed that spironolactone may prevent acute lung injury in COVID-19 infection due to its pleiotropic effects with favorable renin–angiotensin–aldosterone system (RAAS) and ACE2 expression, reduction in transmembrane serine protease 2 (TMPRSS2) activity and antiandrogenic action, and therefore it may prove to act as additional protection for patients at highest risk of severe pneumonia. Cadegiani et al. identified four major factors associated with worse prognosis in COVID-19: age, obesity, hypertension and exposure to androgen hormones, with male sex and androgen-related hair loss that may be associated with enhanced TMPRSS2 expression [6]. Therefore, researchers and clinicians hypothesized that, through its complex pharmacological actions, spironolactone may have therapeutic benefits for patients during the COVID-19 pandemic.

## 2. Aldosterone Action and Mineralocorticoid Receptor Activation

Aldosterone, physiologically, is a hormone of the adrenal cortex, a component of the renin–angiotensin–aldosterone system (RAAS) and a physiological MR activator, which binds to mineralocorticoid receptors (MRs) in the distal tubules and the collecting duct of the kidney, causing sodium reabsorption and secretion of potassium. Mineralocorticoid receptor (MR) activation is a contributing factor to the pathophysiology of many diseases [7]. It has a physiological effect on the regulation of the electrolyte and water balance in the distal tubules, thus maintaining blood pressure and extracellular fluid homeostasis, which acts mainly through cytosolic MRs in epithelial cells [8]. Additionally, stimulation of this system causes vascular stiffness and remodeling, as well as intensification of cardiac inflammation, fibrosis and remodeling [9].

Aldosterone is partly responsible for the increase in extracellular matrix turnover, which is seen in pulmonary, heart and kidney fibrosis and primarily affects the lung epithelium [10]. It is known that elevated aldosterone levels can induce hypertension, alter inflammation and fibrosis and exacerbate cardiovascular disease [11].

Aldosterone also exerts multiple actions on immune cells, which express the mineralocorticoid receptor [7]. It has been shown that activation of MR in immune cells promotes the hyperinflammatory response. In macrophages, MR activation causes polarization towards the M1 proinflammatory phenotype (M1Mϕ). In CD4+ lymphocytes, activation of the MR favors differentiation towards proinflammatory Th17 cells, while enhancing Th17-mediated immunity influences dendritic cells’ (DCs) functioning, crucial for immunological tolerance and homeostasis [12]. It also induces cytotoxic IFNγ+-CD8+ T lymphocytes [7]. This is particularly important in the light of COVID-19 infection characterized by a cytokine storm and hyperinflammatory state, with Th17 T cells increased and increased CD8+ cells cytotoxicity [13].

## 3. Mineralocorticoid Receptor Antagonist Action

Experimental evidence and a literature review suggest that mineralocorticosteroid receptor (MRA) antagonists have beneficial effects on collagen metabolism [10]. Studies have shown that MRAs are effective anti-hypertensive drugs and reduce morbidity and mortality in patients with heart failure with a reduced ejection fraction (HFrEF). The benefits of MRAs may come from diuresis, potassium retention, autonomic effects and reduction in preload and afterload. However, clinical trials in patients with HFrEF or heart failure with a preserved ejection fraction (HFpEF) also suggest that administration of MRAs often causes changes in serum levels of markers of collagen turnover in a way that promotes decreased synthesis [11]. This may reflect both the direct systemic effects of MRAs on collagen metabolism and the inhibitory or reversal of the effects of MRAs on myocardial, vascular and renal fibrosis.

Disturbed RAAS signaling with increased aldosterone-mediated MR activation could constitute an important association between SARS-CoV-2/ACE2 interaction and pneumonia. This may suggest a promising therapeutic option for RAS inhibitors, especially MR antagonists [14]. It is important that, unlike other RAS inhibitors, spironolactone, being an MR antagonist, also has a significant antiandrogenic effect [9]. Such effects may be useful in SARS-CoV-2 infection by inhibiting the precise androgen-dependent expression of TMPRSS2, a transmembrane protease. It is very important for viral entry as it has a stimulating effect on the viral S protein [5,13].

## 4. Pharmacology of Mineralocorticoid Receptor Antagonists

Spironolactone, along with its active metabolites canrenone and eplerenone, belongs to the group of mineralocorticoid receptor antagonists. Additionally, it is a non-selective antagonist that can bind to androgen and progesterone receptors. Spironolactone acts in particular by competitively blocking the action mediated by the aldosterone receptor. The effect of a blockade of this receptor is the blockade of sodium reabsorption with water retention with simultaneous increased potassium retention [15].

The metabolism of spironolactone is multistage. Initially, spironolactone is deacetylated to 7α-thiospironolactone. 7α-thiospironolactone is S-methylated to 7α-thiomethylspironolactone or undergoes elimination to canrenone. Then, 7α-thiomethylspironolactone is reduced to 3α-hydroxytiomethylspironolactone or 3β-hydroxytiomethylspironolactone. Initial studies have shown that canrenone is the major circulating metabolite; however, more recent reports indicate that 7α-thiomethylspironolactone is actually the major metabolite. The route of elimination of spironolactone metabolites is 42–56% in the urine and 14.2–14.6% in the feces. There is no unmetabolized spironolactone in the urine [16].

Potassium canrenoate is the only anti-mineralocorticoid drug available for parenteral administration and has been recommended in patients unable to use oral formulas. It is a prodrug, similarly to spironolactone, metabolized into active canrenone after intravenous infusion (Figure 1).

## 5. Registered MR Antagonist Use in Heart Failure

The mineralocorticoid receptor antagonists (spironolactone and its active metabolites canrenone and eplerenone, potassium canrenoate) competitively block aldosterone-binding receptors. As a result, aldosterone-stimulated protein production is reduced, reducing sodium and water reabsorption and inhibiting urinary excretion of potassium, magnesium and hydrogen cations. The resulting diuretic effect is beneficial in reducing the symptoms of congestion in patients with heart failure (Figure 2).

Spironolactone is an FDA-approved drug for the treatment of heart failure with reduced ejection fraction (HFrEF), refractory hypertension, primary hyperaldosteronism, edema secondary to cirrhosis, edema secondary to nephrotic syndrome that is not adequately controlled by alternative treatments and hypokalemia [17].

Randomized clinical trials (i.e., RALES) have shown both a reduction in mortality and a reduction in the frequency of hospitalizations due to heart failure [11,18]. Improvement in exercise capacity has been demonstrated in patients with features and/or symptoms of stasis [19,20]. On this basis, the guidelines of the European Society of Cardiology (ESC) and the Heart Failure Association strongly recommend (the strength of IB recommendations) the chronic use of spironolactone or eplerenone in all symptomatic patients with heart failure with a reduced ejection fraction [21]. In contrast, in asymptomatic patients with euvolemia or hypovolemia, diuretics may be temporarily suspended. The 2017 American College of Cardiology Guidelines for Heart Failure indicated spironolactone as a drug in NYHA II-IV HFrEF patients with creatinine clearance greater than 30 mL/min and serum potassium levels below 5 mEq/L [22]. There is evidence that spironolactone may reduce myocardial fibrosis, left ventricular mass and extracellular volume in patients with HFpEF [23].

The use of spironolactone in heart failure with a preserved or with an intermediate left ventricular ejection fraction reduces the frequency of hospitalization for heart failure [24]. In these groups of patients, diuretics are recommended to reduce the severity of the features and symptoms of hyperhydration (strength of IB recommendations). In contrast, such patients have not shown a benefit in reducing mortality from treatment with mineralocorticoid receptor antagonists [19,20].

Treatment of high blood pressure is important in all forms of heart failure. The use of diuretics for this purpose is recommended by the ESC [25]. With regard to exacerbation of chronic heart failure, it is recommended to continue oral therapy with documented efficacy, except in cases of hemodynamic instability (symptomatic hypotension, hypoperfusion, bradycardia), hyperkalemia or severe renal failure. In these cases, the dose of oral medications may be reduced or their administration temporarily suspended until the patient’s condition is stabilized [21].

## 6. Potential MR Antagonist Use in the Treatment of Pulmonary Fibrosis

There are reports that the use of spironolactone may be important in the prevention of pulmonary fibrosis (Figure 3) [4]. The limitations of some of the studies that claim to benefit from spironolactone are related to the fact that they were conducted in animal models such as rats or other rodents [26]. There are also no direct studies to show a beneficial effect of a mineralocorticoid receptor antagonists in pulmonary fibrosis following viral infection.

When factor-induced damage occurs in lung tissue, a set of growth factors and cytokines including monocyte chemoattractant protein-1 (MCP-1), transforming growth factor β1 (TGF-β1), tumor necrosis factor α (TNF-α), interleukin-1β (IL-1β) and interleukin-6 (IL-6) are overexpressed and released by cells [27]. Type II follicular endothelial cells are one of the main sources of these fibrogenic factors. These factors stimulate hyperproliferation of type II follicular cells, recruit fibroblasts for localized fibrosis and induce differentiation and activation of fibroblasts into myofibroblasts. Myofibroblasts are responsible for the excess build-up of ECM in the basement membranes and interstitial tissues, eventually leading to loss of alveolar function, especially gas exchange between alveoli and capillaries [27].

In an experimental model study of Sprague-Dawley rats, Zhou H. and colleagues demonstrated the beneficial effect of an aldosterone antagonist in the treatment of renal fibrosis. In this model, isoprenaline (Iso) was injected subcutaneously to induce heart failure that promoted renal fibrosis. Rats treated with spironolactone were dosed by gavage with spironolactone at a dose of 30 or 60 mg/kg/day for a period of 21 days, after which cardiac and fibrosis indicators were measured. Pathological changes and expression of type I and III collagen, α-smooth muscle actin, differentiation cluster-31 and TGF-β were examined. In this study, rats treated with Iso showed poorer heart function, impaired renal structure and higher levels of renal collagen deposition (i.e., fibrosis) compared to the control group. The level of TGF-β, a key regulating factor of EndMT, increased in the rats treated with Iso. Treatment with spironolactone reduced these effects, so spironolactone may improve renal fibrosis by inhibiting the endothelial–mesenchymal transition (EndMT). Moreover, TGF-β may play a role as one of the key factors. There is also a study that has shown that spironolactone can reduce fibrosis and thus improve renal function by inhibiting the renin–angiotensin system and suppressing TGF-β [28].

Another study also investigated the effect of spironolactone in reducing acute lung injury by administering spironolactone following intestinal ischemia and reperfusion (I/R) to rats in an experimental animal model [4]. Activated polymorphonuclear neutrophils and reactive oxygen species contribute to lung damage caused by intestinal I/R. Spironolactone, an antagonist of mineral corticosteroid receptors, is protective against I/R damage in animal models of the retina, kidneys, heart and brain [29]. In Barut F. et al.’s study, Wistar albino rats were divided into four groups: (1) sham control; (2) intestinal I/R (30 min of ischemia by occlusion of the superior mesenteric artery, followed by 3 h of reperfusion); (3) pretreatment with spironolactone (20 mg/kg) + I/R; and (4) pretreatment with spironolactone without I/R. Spironolactone was administered orally 3 days prior to the I/R of the gut. The lipid peroxidation marker (malondialdehyde; MDA), oxidation index or state (reduced glutathione; GSH), polymorphonuclear neutrophil sequestration index (myeloperoxidase; MPO), inducible nitric oxide synthase (iNOS) and histopathological analysis of lung tissue were studied. These investigators showed that pretreatment with spironolactone significantly reduced the intestinal I/R-induced lung damage as evidenced by the histological results and the levels of MDA and MPO. In addition, pretreatment decreased iNOS (inducible nitric oxide synthase) immunoreactivity [4].

## 7. Potential Pharmacological Actions of Spironolactone in COVID-19

Large-scale epidemiological reports on COVID-19 underlined that, apart from age and co-morbidities, additional risk factors include obesity, hypertension and male gender, all of which have been associated with mineralocorticoid action [30,31,32]. It is possible that spironolactone and its active metabolite (canrenoate), through combined pleiotropic action, i.e., mitigating abnormal ACE2 and TMPRSS2 expression, antiandrogen effect and MR blocking, antifibrotic and anti-hyperinflammatory action, may provide therapeutic benefit to patients with SARS-CoV-2 infection including severe COVID-19 pneumonia.

The presence of the ACE2 receptor and TMPRSS2 on the cell membrane is crucial for viral entry into the cells. The co-expression of these two proteins varies between cells but has been shown to be present at the highest level in nasal cavity type 2 pneumocyte cells [33,34], which is in agreement with the clinical symptoms of this infection. Abnormalities in ACE2 expression in patients with hypertension and obesity [35,36] and abnormalities in TMPRSS2 expression in individuals exposed to high levels of androgens [37,38] have been shown to be risk factors for COVID-19 [6].

MRAs competitively inhibit mineralocorticoid receptors and decrease the number of epithelial sodium channels in the distal renal tubule. Spironolactone, an MRA, has long been used for the treatment of hypertension; however, its non-specificity for mineralocorticoid receptors manifests as antiandrogenic and progestational effects [11]. Aldosterone, the terminal hormone of RAAS, exerts 90% of the mineralocorticoid activity of adrenal secretions and is a key regulator of the sodium, potassium and body fluid balance. Angiotensin II and increased extracellular K^+^ concentration, the strongest secretagogues for aldosterone, increase expression of the CYP11B2 gene, which encodes aldosterone synthase. Acting via the mineralocorticoid receptor (MR), aldosterone modulates the expression of ion channels, pumps and exchangers in epithelial tissues (kidney, colon and salivary and sweat glands). This ultimately leads to an increase in transepithelial Na^+^ and water reabsorption and K^+^ excretion [39]. Mineralocorticoid receptors are also found in non-epithelial tissues such as the retina, brain, myocardium, vascular smooth muscle cells, macrophages, fibroblasts and adipocytes. The effects of aldosterone are therefore widespread, extending well beyond its role as a “renal hormone.” Specifically, aldosterone is thought to mediate inflammation and affect energy metabolism in non-epithelial tissues. The renin–angiotensin–aldosterone system (RAAS) is an important neurohormonal pathway that augments collagen synthesis in the myocardium and systemic vasculature [40,41]. The mineralocorticoid receptor (MR) blocker spironolactone has antifibrotic properties and has shown to be beneficial in patients with systolic left heart failure. Spironolactone is frequently used as a diuretic in PAH [42].

TMPRSS2 is a serine protease that reacts to androgens and also cleaves the SARS-CoV-2 spike protein. This mechanism facilitates the penetration and activation of the virus in the body. TMPRSS2 is expressed in many tissues outside the lung and is well known for its predominant expression in the prostate epithelium and its role in prostate carcinogenesis [43]. The androgen-regulated promoter TMPRSS2 fuses with the coding regions of members of the proto-oncogenic transcription factor family of ETS, which is strongly associated with prostate cancer and regulates many biological processes. TMPRSS2 is also expressed in the endothelium of other organs such as the heart, kidneys and gastrointestinal tract, which may suggest that they could be important target organs for SARS-CoV-2 infection [44]. All of the potential actions are presented in Figure 4.

### 7.1. Antiandrogen Action

Current evidence indicates that the antiandrogen activity of spironolactone decreases the expression of TMPRSS2 and exhibits concurrent actions in the modulation of ACE2 expression, thus impeding SARS-CoV-2 entry into the cells [6,45]. Another action is to increase the levels of angiotensin 1–7 which counterbalance the angiotensin II-AT-1 axis overexpression [45].

Epidemiological data suggest that there does not seem to be a difference in the frequency of cases of COVID-19 infection between sexes, but men are about twice as likely to die from COVID-19 than women [46,47,48]. It is still not clear whether this difference is not attributed to lifestyle. Nevertheless, the data also suggest that androgens may be associated with SARS-CoV-2 pathogenesis. Moreover, preliminary observation of the high frequency of male pattern hair loss among admitted COVID-19 patients suggested that androgen levels might be associated with the severity COVID-19 infection [32]. Wambier et al. analyzed a series of 175 patients with confirmed SARS-CoV-2 infection, 122 males and 53 females, and reported that 67% of patients with COVID-19 presented with clinically relevant androgenetic alopecia (AGA) [49]. In line with this observation, it has been suggested that pre-pubertal children who have not been exposed to androgens seem to have a milder course of SARS-CoV-2 infection; however, this may also be due to other factors, as young people are usually healthy and have not yet made bad lifestyle choices [50]. McCoy et al. proposed the concept that there is a relationship between the amount of testosterone and the severity of the condition in COVID-19 [51]. The statistical analysis of the course of the disease showed an almost 6–16 times higher number of deaths in patients with African American ethnicity compared to other patients [52]. This may be related to the increased sensitivity to androgens, which is correlated with, among others, prostate cancer and androgenetic alopecia [51].

The androgen receptor (AR) has been shown to be expressed in murine type II pneumocytes, and, more importantly, androgen administration in murine models significantly altered lung gene expression [53].

Taken together, it is plausible that that androgen receptor (AR) may be involved in modulation of the patient’s response to COVID-19 and patients may benefit from antiandrogen treatment. The administration of antiandrogen treatments in COVID-19 patients has been suggested by some authors [54], and, moreover, therapeutic randomized controlled clinical trials with degarelix (NCT04397718), bicalutamide (NCT04374279) and spironolactone (NCT04345887) have been initiated.

### 7.2. Anti-Hyperinflammatory Action

The development of a hyperinflammatory state in COVID-19 may trigger ARDS. This is particularly important in the light of COVID-19 infection characterized by a cytokine storm and hyperinflammatory state, with Th17 T cells increased and increased CD8+ cells cytotoxicity [13,55]. Sites of potential pharmacological actions of MR activation in COVID-19 include reduction in polarization towards the M1 proinflammatory phenotype (M1Mϕ) in macrophages, reduction in proinflammatory Th17 CD4+ lymphocytes and reduction in cytotoxic IFNγ+-CD8+ T lymphocytes [7].

Moreover, spironolactone is known for its potential to reduce TNF-α and MCP-1 in monocytes [56] and may cause a significant reduction in proinflammatory cytokines, such as reduced levels of IFNγ and TNF-α as well as decreased gene transcription for many regulators of inflammation [57]. A suppressive effect was also demonstrated on IL-2, IL-6, IL-15 and granulocyte-macrophage colony stimulating factor (GM-CSF) [58,59]. Impaired neutrophil recruitment and host defenses against SARS-COV-2 may also be caused by the registered reduced number of IL-17 and G-CSF (granulocyte colony-stimulating factor) transcripts [59].

Recently, also spironolactone in combination with vitamin D3 has been shown to inhibit NF-κB activity in mouse macrophages and further decrease the expression of proinflammatory cytokines in a skin injury model [60]. Importantly, the inhibition of NF-κB by spironolactone was shown to be independent of the mineralocorticoid receptor [61].

The level of cytokines such as interleukin (IL)-6, IL-1β and tumor necrosis factor-α has been shown to be higher in those with severe symptoms of COVID-19 infection [62] and, as described in the above findings, may indicate the anti-inflammatory activity of spironolactone.

### 7.3. Impact on Apoptosis of Immune Cells

It was also shown that spironolactone affects the expression of apoptosis-related genes, leading to late apoptosis of blood mononuclear cells (MNC) at even relatively low concentrations [59]. Moreover, spironolactone was associated with induction of cytokine production suppression in MNC cultures that led to apoptosis [63]. Very recently, spironolactone was also shown to inhibit DNA damage response mechanisms in cancer stem cells, which further indicates a proapoptotic activity mode of action of this compound [64]. Taken together, modulation of the apoptotic response of the cells may be critical for the immune response and subsequent outcomes of the COVID-19 infection, but this hypothesis needs further investigation.

### 7.4. Antifibrotic Action

According to the available knowledge, the use of spironolactone may have positive effects in the prevention of fibrosis [4,65,66]. The spironolactone-dependent up-regulation of the adenosine A2A receptor (A2AR) has recently been shown to play a role in endothelial–mesenchymal transition, suggesting a possible mechanism for spironolactone in the reduction in fibrosis [67]. Moreover, as mineralocorticoid receptors (MR) have already been proposed as “master regulators of extracellular matrix remodeling”[68], it is likely that spironolactone may modulate the extracellular matrix and fibrosis via interaction with this receptor. Furthermore, the most common symptoms of elevated aldosterone levels are hypertension and exacerbation of cardiovascular disease, but in addition, they can modify inflammation and contribute to fibrosis [65], and spironolactone is likely to compensate for the effects of the elevated aldosterone levels. Lieber et al. showed that the use of spironolactone alleviated pneumonia induced by liposaccharides and bleomycin by reducing the number of inflammatory cells such as lymphocytes, neutrophils, macrophages and eosinophils in the alveoli [69]. Ji et al. demonstrated the therapeutic effect of spironolactone by reducing the inflammatory response in the lungs [70]. Atalay et al. demonstrated a positive effect of spironolactone in the treatment of acute lung injury [71], while Barut et al., in a study aimed at evaluating the effect of spironolactone on lung damage caused by intestinal ischemia and reperfusion, demonstrated reduced neutrophil infiltration, oxidative stress and histopathological damage [4]. Overall, the effect of aldosterone on the pathophysiology and incidence of fibrosis is not fully understood, most studies of these effects of aldosterone have been conducted in animal models and, similarly, there is no evidence that the use of aldosterone antagonists may produce positive effects in pulmonary fibrosis. Nevertheless, considering the above findings, it seems likely that spironolactone may be effective in the prevention of COVID-19 infection-related pulmonary fibrosis, but further experimental research is needed to elaborate the mechanisms of involvement of this medication in the pathology of pulmonary fibrosis [3].

### 7.5. Oxidative Stress

SARS-CoV-2, like other RNA viruses, has the potential to trigger oxidative stress, causing an imbalance in the cellular redox environment, which can result in the production of reactive oxygen species and antioxidant defenses [72,73]. This is especially important in patients exposed to high fractions of inspired oxygen during mechanical ventilation or noninvasive ventilation high-flow oxygen therapy [74]. It has been shown that spironolactone exhibits antioxidant activity and protects organs against oxidative stress by inhibiting the production of free radicals [69]. There is also a substantial amount of evidence from animal models that spironolactone can play a role in the reduction in oxidative stress and antioxidant defenses, e.g., in hyperthyroid rats [75]. Moreover, spironolactone has long been shown to inhibit aldosterone-induced oxidative stress in human endothelial cells [76]. The reduction in oxidative stress may have a significant effect on the outcomes of COVID-19 infection, especially in patients with severe outcomes.

## 8. Conclusions

Spironolactone and potassium canrenoate, exerting combined pleiotropic action, may provide a therapeutic benefit to patients with COVID-19 pneumonia through antiandrogen, MR blocking, antifibrotic and anti-hyperinflammatory action. Future prospective clinical trials are warranted to evaluate its therapeutic potential. The potential benefit, however, may be unacceptable to many male patients due to the antiandrogen action of spironolactone and cannot be dismissed. Clinical trials aimed at treating fibrosis after SARS-CoV-2 infection with spironolactone and potassium canrenoate are ongoing worldwide. Moreover, spironolactone exerts a potential effect early during SARS-CoV-2 viral replication and spread; therefore, it seems to be an ideal candidate drug not only for combating the severe and long-term complications of severe SARS-CoV-2 infection, but also for the prophylactic and early treatment of COVID-19.

## Figures and Tables

**Figure 1 pharmaceuticals-14-00071-f001:**
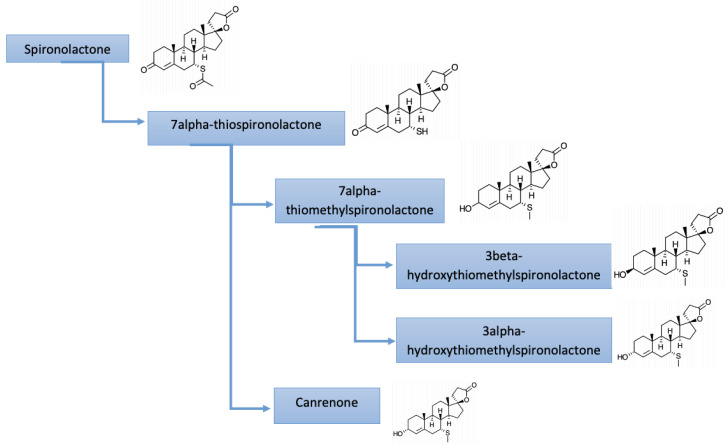
Metabolites of spironolactone.

**Figure 2 pharmaceuticals-14-00071-f002:**
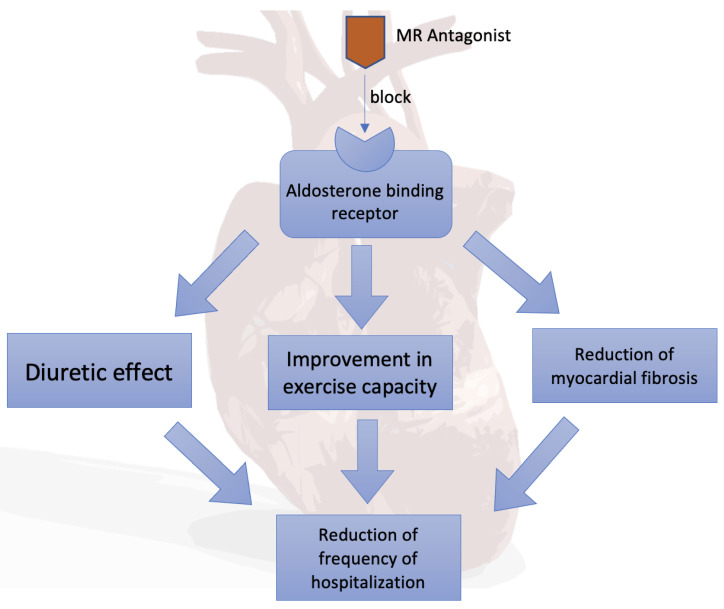
Mineralocorticoid receptor (MR) antagonists in heart failure.

**Figure 3 pharmaceuticals-14-00071-f003:**
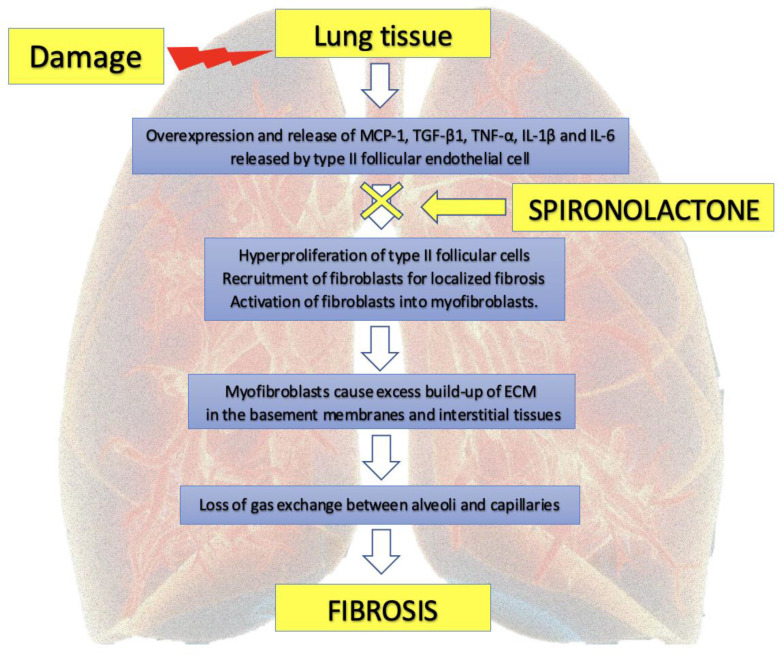
Potential effect of spironolactone in prevention of pulmonary fibrosis. MCP-1: monocyte chemoattractant protein-1; TGF-β1: transforming growth factor beta-1; TNF-α: tumor necrosis factor alfa, IL-1β: interleukin-1b; IL-6: interleukin-6; ECM: extracellular matrix.

**Figure 4 pharmaceuticals-14-00071-f004:**
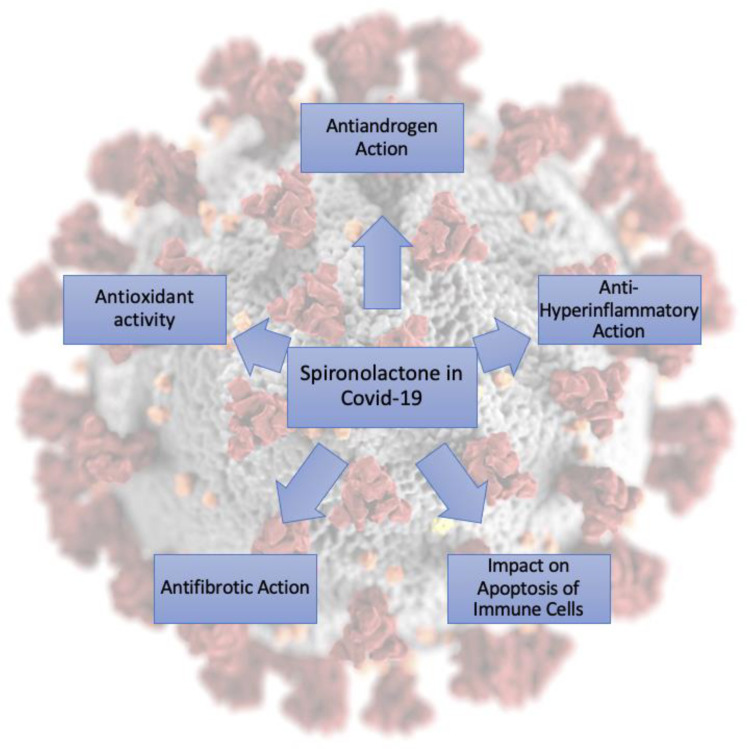
Potential pharmacological actions of spironolactone in COVID-19.

## Data Availability

The data presented in this study are available in the main text.

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
