# Peer review of "COVID-19—The Potential Beneficial Therapeutic Effects of Spironolactone during SARS-CoV-2 Infection"

_pharmaceuticals, 2021, doi:10.3390/ph14010071_

Round 1

Reviewer 1 Report

This manuscript claims the hypothesis of spironolactone's potentials for treating COVID-19. Here are my comments;

  1. MR antagonist property is mostly related to treating heart failure and pulmonary fibrosis. Besides, authors states MR antagonist has not been proven to beneficial effect in pulmonary fibrosis following viral infection (line 173-175), so it seems logically weak to combine with COVID-19 treatment.
  2. More detailed statement regarding spironolactone's beneficial effects on RAAS and TMPRSS2 activity is required.
  3. Spironolactone seems effective on the severe cases of COVID-19 because of anti-androgenic and anti-inflammatory property. I'd like to request authors' opinion from the view point of clinical application.

Author Response

Thank you for giving us the opportunity to provide the revised version of the manuscript pharmaceuticals-1037142 titled: ”COVID-19 - The potential beneficial therapeutic effects of spironolactone during SARS-CoV-2 infection”. I hereby submit this revision on behalf of all of the co-authors. Thank you for the helpful critique of our manuscript. As suggested, we have responded to each query and comment made by the Reviewers and the Editor in a point-by-point response letter. In this letter and the revised manuscript, we have included the exact wording of each query and our response, formatted as requested in your letter.

Reviewer 1:
This manuscript claims the hypothesis of spironolactone's potentials for treating COVID-19. Here are my comments;

  1. MR antagonist property is mostly related to treating heart failure and pulmonary fibrosis. Besides, authors states MR antagonist has not been proven to beneficial effect in pulmonary fibrosis following viral infection (line 173-175), so it seems logically weak to combine with COVID-19 treatment.

Response: Thank you for this comment. The presented article is a hypothesis. The use of spironolactone has been proven to prevent fibrosis in conditions other than viral infections. This is why we present this article as an introduction to scientific research. A clinical trial from our center is currently underway (EudraCT nr 2020-004134-38), along with other studies NCT04345887, NCT04643691, NCT04424134.

  1. More detailed statement regarding spironolactone's beneficial effects on RAAS and TMPRSS2 activity is required.

Response: Thank you for this suggestion. We’ve added more details (lines 234-261).

  1. Spironolactone seems effective on the severe cases of COVID-19 because of anti-androgenic and anti-inflammatory property. I'd like to request authors' opinion from the view point of clinical application.

Response: Thank you for this question. There is currently no single working drug that we can use in patients infected with Covid-19. The profile of spironolactone activity suggests a positive effect on the course of infection, however, there are no completed studies, especially randomized ones, that would explain the mechanism of spironolactone action. Improvement in the health of critically ill patients may be related to spironolactone use, and attempts to find the "perfect drug" often result in over-excitement. To the best of our knowledge, there are currently 3 clinical trials registered on the ClinicalTrials.gov website testing the effects of spironolactone and one registered with EudraCT.

Due to the lack of certain results and reliance on premises and individual cases, we did not include a clear answer to the question in the text, which does not mean that it is not worth trying to use this drug in patients with Covid-19.

Reviewer 2 Report

This review by Kotfis et al concerns the potential therapeutic use of spironolactone, an anti-hypertensive and anti-androgenic medication. This topic is of extremely high interest in the context of the ongoing global pandemic. The concept is not exactly new, with numerous full and abbreviated reviews and commentaries already discussing the potential for therapeutic use of this precise drug for this precise indication. The authors, do however, thoroughly contextualize the use of spironolactone towards indication in COVID-19 with its prior use as an anti-fibrotic drug and anti-hyperinflammatory action. This is the key point which sets this review apart from the rest, which do not spend as much time discussing such points. Improvement can be made in the presentation of this review.

1) The authors cover a number of highly complex, but related topics. However, there is almost a complete absence of figures to help guide the authors through these complex topics. The authors are strongly encouraged to include figures to help illustrate the important and well-described points made in sections 5, 6 and 7.

2) The only figure which is included involves a brief description of the metabolite byproducts of spironolactone. The points made about metabolism of spironolactone are highly superficial and the reason for their discussion is not very obvious. The figure is very basic and doesn't do much more than a simple table could. Also, typo lines (the red squiggle lines under the words) are kept in the figure, which should be removed.

Author Response

Thank you for giving us the opportunity to provide the revised version of the manuscript pharmaceuticals-1037142 titled: ”COVID-19 - The potential beneficial therapeutic effects of spironolactone during SARS-CoV-2 infection”. I hereby submit this revision on behalf of all of the co-authors. Thank you for the helpful critique of our manuscript. As suggested, we have responded to each query and comment made by the Reviewers and the Editor in a point-by-point response letter. In this letter and the revised manuscript, we have included the exact wording of each query and our response, formatted as requested in your letter.

Reviewer 2
This review by Kotfis et al concerns the potential therapeutic use of spironolactone, an anti-hypertensive and anti-androgenic medication. This topic is of extremely high interest in the context of the ongoing global pandemic. The concept is not exactly new, with numerous full and abbreviated reviews and commentaries already discussing the potential for therapeutic use of this precise drug for this precise indication. The authors, do however, thoroughly contextualize the use of spironolactone towards indication in COVID-19 with its prior use as an anti-fibrotic drug and anti-hyperinflammatory action. This is the key point which sets this review apart from the rest, which do not spend as much time discussing such points. Improvement can be made in the presentation of this review.

Response: Thank you for these kind comments.

1) The authors cover a number of highly complex, but related topics. However, there is almost a complete absence of figures to help guide the authors through these complex topics. The authors are strongly encouraged to include figures to help illustrate the important and well-described points made in sections 5, 6 and 7.

Response: Thank you for this suggestion. We have included Figures 2-4 in the publication. They are based on the text in chapters 5,6,7, but in an attempt to maintain readability, they are only a visualization of the main points. All details are intentionally omitted and left in the text.

2) The only figure which is included involves a brief description of the metabolite byproducts of spironolactone. The points made about metabolism of spironolactone are highly superficial and the reason for their discussion is not very obvious. The figure is very basic and doesn't do much more than a simple table could. Also, typo lines (the red squiggle lines under the words) are kept in the figure, which should be removed.

Response: Thank you for this suggestion. We’ve corrected typo lines.

Moreover, we added two introductory sentences to section 7.5 as they were omitted accidentally in the previous version. Thank you once again for the opportunity to resubmit this manuscript, and we hope you find it suitable for publication in the Pharmaceuticals.

With best regards

Katarzyna Kotfis, MD, PhD, DESA

Round 2

Reviewer 1 Report

Manuscript is acceptable for publication.

Reviewer 2 Report

The authors have satisfactorily addressed my concerns. I have no further comment.